# Zearalenone Promotes Uterine Hypertrophy through AMPK/mTOR Mediated Autophagy

**DOI:** 10.3390/toxins16020073

**Published:** 2024-02-01

**Authors:** Lijie Yang, Wenshuang Liao, Jiuyuan Dong, Xiangjin Chen, Libo Huang, Weiren Yang, Shuzhen Jiang

**Affiliations:** Key Laboratory of Efficient Utilization of Non-Grain Feed Resources (Co-construction by Ministry and Province), Ministry of Agriculture and Rural Affairs, College of Animal Sciences and Veterinary Medicine, Shandong Agricultural University, Tai’an 271018, China; yanglijie@sdau.edu.cn (L.Y.); liaowenshuang0408@163.com (W.L.); dongjiuyuan91@163.com (J.D.); chenxiangjin2002@163.com (X.C.); huanglibo@sdau.edu.cn (L.H.); wryang@sdau.edu.cn (W.Y.)

**Keywords:** mycotoxin, weaned piglets, proliferation, apoptosis, uterine endometrial epithelium cells

## Abstract

Zearalenone (ZEN), a non-steroidal *Fusarium graminearum* with an estrogen effect, can cause damage to the gastrointestinal tract, immune organs, liver, and reproductive system. Further analysis of the mechanism of ZEN has become an important scientific issue. We have established in vivo and in vitro models of ZEN intervention, used AMPK/mTOR as a targeted pathway for ZEN reproductive toxicity, and explored the molecular mechanism by which ZEN may induce uterine hypertrophy in weaned piglets. Our study strongly suggested that ZEN can activate the phosphorylation of AMPK in uterine endometrial epithelium cells, affect the phosphorylation level of mTOR through TSC2 and Rheb, induce autophagy, upregulate the expression of proliferative genes PCNA and BCL2, downregulate the expression of apoptotic gene BAX, promote uterine endometrial epithelium cells proliferation, and ultimately lead to thickening of the endometrial and myometrium, increased density of uterine glands, and induce uterine hypertrophy.

## 1. Introduction

Reproductive disorder has become one of the important diseases restricting the high yield of sows. The direct economic losses caused by non-communicable sow reproductive disorders caused by feed mold worldwide each year exceed 100 trillion yuan [1]. ZEN is one of the most common mycotoxins in various cereals of corn, barley, wheat, and rice, which is mainly produced by *Fusarium graminearum* [2,3,4]. ZEN and its metabolites are absorbed through the intestinal mucosa, enter the liver and blood circulation through the portal vein, and are transported to various organs [5,6,7,8]. Normally, ZEA exhibits white particles with good thermal stability and is soluble in chloroform, alcohols, and some alkaline solutions [4]. Studies have shown that ZEN can competitively bind to estrogen receptors (ERs) to induce abnormal uterine development, vaginitis, and infertility, leading to other reproductive problems such as precocious puberty, infertility, and abortion [9,10,11,12]. Different species have unique sensitivities to ZEN, and among these, pigs are the most sensitive species [13]. In females, ZEN can cause abnormal estrogen levels, change reproductive organ morphology and function, and inhibit follicular maturation, thereby reducing oocyte number, reducing fertility, and increasing embryonic lethality and abortion. In male animals, ZEN can cause testicular atrophy, decreased androgen secretion, increased proportion of abnormal sperm, lower post-mating pregnancy rate, and reduced acrosome reaction and the ability of sperm to bind to zona pellucida [4,5,7].

Autophagy is a process that degrades misfolded proteins, dead organelles, and pathogens to ensure cell development [14,15]. When the body is under stress, AMP/ATP changes and activated AMPK initiates autophagy through coordinated activation of ULK 1 and negatively regulated mTOR [11,16,17,18]. Cell proliferation and apoptosis are self-repair mechanisms [19]. During apoptosis, internucleosomal DNA enters the vesicle and is engulfed by other cells [20]. Apoptotic cells are immediately eliminated by cells with phagocytic activity, avoiding the autoimmune response of the cells [21] and facilitating the stability of the tissue environment [22]. Thus, deep analysis of the mechanism of ZEN has become an important scientific problem. Our study aims to establish in vivo and in vitro models of ZEN intervention, using AMPK/mTOR as a targeted pathway for ZEN reproductive toxicity, and explore the molecular mechanism by which ZEN may induce uterine hypertrophy in weaned piglets.

## 2. Results

### 2.1. ZEN-Induced Uterine Hypertrophy in Weaned Piglets

A dietary ZEN of 1.5 to 3.0 mg/kg feed resulted in uterine hyperemia and enlargement of weaned piglets (Figure 1A). The results of a uterine histomorphology showed that ZEN increased the number (e-f-g-h) and length (m-n-o-p) of glands in the uterus and increased the number of monolayers of cubic epithelial cells in the endometrium (i-j-k-l, red arrow). Bilayer nuclei were observed in the endometrial epithelium (l) and endometrial glands (p) of the ZEN 3.0 group (Figure 1B). With the increase in dietary ZEN, the thickness of the myometrium and endometrium were linearly increased (*p* < 0.05), and there were significant differences among groups (*p* < 0.05, Figure 1C).

### 2.2. ZEN-Induced Endometrial Epithelial Cells to Be Active

The result of the ultrastructure of uterine endometrial epithelium showed that the nucleolus is obvious, the nuclear membrane boundary and mitochondrial inner spine are clear in the control group (Figure 2; Control A~C; red arrow), and there is vacuolation of the endoplasmic reticulum occasionally (yellow arrow). In the treatment group, the number of nucleoli increased, and the boundaries were blurred (ZEN 0.15 A~C, ZEN 1.5 A~C, ZEN 3.0 A~C; red arrows). Under 25,000 and 50,000 times of microscopy, the endoplasmic reticulum was seriously vacuolated, and the organelle structure was damaged irregularly (ZEN 0.15 D~E, ZEN 1.5 D~E, ZEN 3.0 D~E; yellow arrows). In addition, we found that cells in the ZEN 0.15 and ZEN 1.5 groups had multiple nucleoli, while the ZEN 3.0 group had the largest number of cells with multiple nucleoli.

### 2.3. The Distribution of AMPK/mTOR Pathway-Related Genes in the Uterus

The results of the immunohistochemical (IHC) staining indicated that the *p*-AMPK and *p*-mTOR immunoreactivity was mainly localized in the endometrium, uterine glands, and endometrial epithelial cells (Figure 3 and Figure 4). As shown by the red arrow, the distribution of positive reactions was consistent for each treatment, and no significant difference was observed. As for *p*-AMPK, the control group had fewer immune-positive reactive substances. As the ZEA increased, the *p*-AMPK positive reaction gradually increased (Figure 3; A1~2, B1~2, C1~2, and D1~2). With the increase of ZEA in the diet, the color of *p*-AMPK-positive substances in endometrial epithelial cells (A3, B3, C3, and D3) and uterine gland cells (A4, B4, C4, and D4) gradually deepened. In terms of *p*-mTOR, the control group showed significant immunoreactive substances, and as the ZEA increased, the positive reactions gradually weakened (Figure 4; A1~2, B1~2, C1~2, and D1~2), the color of positive reaction substances in endometrial epithelial cells (A3, B3, C3 and D3) and endometrial gland cells (A4, B4, C4 and D4) gradually became lighter.

### 2.4. The Distribution of LC3 and Beclin1 in the Uterus

The immunohistochemical positive substances of microtubule-associated protein 1 light chain 3 alpha (LC3) and Beclin1 are yellow, mainly distributed in the endometrial layer, uterine glands, and uterine endometrial epithelium (Figure 5 and Figure 6). The distribution of positive substances in each treatment was consistent, and no significant difference was observed (red arrow). There were fewer immunoreactive substances in the control group, and the positive reactions gradually increased with the increase in ZEN (LC3, Figure 5, A1~2, B1~2, C1~2 and D1~2; Beclin1, Figure 6, A1~2, B1~2, C1~2 and D1~2;), the color of positive reactive substances in endometrial epithelial cells (A3, B3, C3 and D3) and uterine glandular cells (A4, B4, C4 and D4) gradually deepened.

### 2.5. The Distribution of PCNA in the Uterus

The positive reactive substance of proliferating cell nuclear antigen (PCNA) immunohistochemistry is yellow, and the positive reactive substance is mainly distributed in the endometrial layer, uterine glands, and uterine endometrial epithelium (Figure 7). The distribution of positive reactive substances in each treatment is consistent, and there is no significant difference (red arrow). The control group had fewer immune positive reactive substances, and as the ZEN increased, the PCNA positive reaction gradually increased (A1~2, B1~2, C1~2, and D1~2), the color of PCNA positive substances in endometrial epithelial cells (A3, B3, C3, and D3) and uterine glandular cells (A4, B4, C4, and D4) gradually deepened.

### 2.6. The Expression of Genes Involved in Autophagy, Proliferation, and Apoptosis

With the increase in ZEN, the relative mRNA expression of Beclin1, LC3, ATG5, ATG7, and ATG9 increased linearly (*p* < 0.05, Figure 8). There was a significant difference (*p* < 0.05) in the relative mRNA expression of ATG5 among the groups. In terms of Beclin1, LC3, ATG7, and ATG9, significant differences (*p* < 0.05) were observed between the ZEN treatment (0.15~3.0 mg/kg feed) groups, while there was no statistical difference (*p* > 0.05) between the ZEN0.15 and control groups. In addition, with the increase in ZEN, the relative mRNA expression of PCNA and BCL2 increased linearly (*p* < 0.05), while BAX decreased linearly (*p* < 0.05). There was a significant difference (*p* < 0.05) between the ZEN treatment (0.15~3.0 mg/kg feed) groups, while there was no statistical difference (*p* > 0.05) between the ZEN0.15 and the control groups. In addition, we also found that the content of ZEN, α-zearalenone, and β-zearalenone was significantly positively correlated with Beclin1, LC3, ATG5, ATG7, ATG9, PCNA, and BCL2 (*p* < 0.01), while negatively correlated with BAX (*p* < 0.01, Appendix A). The autophagy genes Beclin1, LC3, ATG5, ATG7, and ATG9 were significantly positively correlated with the proliferation genes PCNA and BCL2 (*p* < 0.01), while they were significantly negatively correlated with the apoptosis gene BAX (*p* < 0.01, Appendix A). The relative protein expression of *p*-AMPK/AMPK increased linearly with the increase in dietary ZEN (*p* < 0.05), whereas the *p*-mTOR/mTOR decreased linearly (*p* < 0.05, Figure 9). Significant differences (*p* < 0.05) were observed between the ZEN treatment (0.15~3.0 mg/kg feed) groups, while there was no statistical difference (*p* > 0.05) between the ZEN0.15 and control groups. As for autophagy genes, the relative protein expressions of Beclin1, LC3Ⅱ/Ⅰ, ATG5, ATG7, and ATG9 were linearly increased with the increase in dietary ZEN (*p* < 0.05). Also, there was a significant difference (*p* < 0.05) between the ZEN treatment (0.15~3.0 mg/kg feed) groups, while there was no statistical difference (*p* > 0.05) between the ZEN0.15 and the control groups. In terms of proliferation and apoptosis genes, as the concentration of ZEN increases, the relative expression levels of PCNA and BCL2 proteins increase linearly (*p* < 0.05), while the BAX decreases linearly (*p* < 0.05). Significant differences (*p* < 0.05) were observed between the ZEN treatment (0.15~3.0 mg/kg feed) groups, while there was no statistical difference (*p* > 0.05) between the ZEN0.15 and control groups.

### 2.7. Transcriptome Analysis of Endometrial Epithelial Cells

After treating pig endometrial epithelial cells with ZEN (0, 5, 10, 20, 40 and 80 μmol/L) for 24 h, 5 μmol/L ZEN increased cell viability by 10.32%; 20, 40, and 80 μmol/L ZEN decreased cell viability by 11.06%, 33.41% and 46.28%, respectively, while 10 μmol/L ZEN had no significant effect (Appendix A). After 48 h of treatment, we found that 10, 20, 40, and 80 μmol/L ZEN reduced cell viability by 23.83%, 39.87%, 59.01%, and 75.74%, respectively (Appendix A). Through the transcriptome analysis of endometrial epithelial cells, we identified 55 genes related to the AMPK-mTOR pathway (Figure 10A). Compared with the control group, the ZEN5 group significantly upregulated seven genes (*p* > 0.05) and downregulated eight genes. The ZEN20 group significantly upregulated 17 genes (*p* < 0.05) and downregulated 14 genes (*p* < 0.05). The ZEN40 group significantly upregulated 25 genes (*p* < 0.05) and downregulated 21 genes (*p* < 0.05).

### 2.8. Validation of Relative mRNA and Protein Expression of Candidate Genes

With the increase in ZEN (0~40 μmol/L), the mRNA relative expression of ULK1, TSC2, Beclin1, LC3, ATG5, ATG7, ATG9, and BAX in endometrial epithelial cells were linearly increased (*p* < 0.05, Appendix A). However, the mRNA relative expression levels of BCL2 and Rheb were linearly decreased (*p* < 0.05). In addition, we also found that the mRNA relative expressions of ULK1, TSC2, Rheb, Beclin1, LC3, ATG5, ATG7, and ATG9 were significantly different (*p* < 0.05) among ZEN treatments (5–40 μmol/L). There was no significant difference (*p* > 0.05) between the ZEN5 and control group. In terms of the mRNA expression of PCNA and BCL2, the order from high to low was ZEN5 > Control > ZEN20 > ZEN40 (*p* < 0.05), while the relative mRNA expressions of BAX were ZEN5 < Control < ZEN20 < ZEN40 (*p* < 0.05).

The results of relative protein expression showed that there were significant differences in the relative protein expression levels of *p*-AMPK/AMPK, *p*-mTOR/mTOR, PCNA, and Beclin1 among different ZEN treatment groups (*p* < 0.05, 5–40 μmol/L). In terms of *p*-AMPK/AMPK and Beclin1, ZEN significantly increased their relative protein expression (*p* < 0.05), but there was no statistical difference between the ZEN5 and the control group (*p* > 0.05). In terms of *p*-mTOR/mTOR, ZEN treatment significantly reduced its protein relative expression (*p* < 0.05), but there was also no statistical difference between the ZEN5 and the control group (*p* > 0.05). As for PCNA, the relative protein expression ZEN5 was significantly higher than the control group (*p* < 0.05), while the control group was significantly higher than the ZEN20 and ZEN40 groups (*p* < 0.05).

## 3. Discussion

### 3.1. ZEN-Induced Uterine Hypertrophy and Endometrial Epithelial Cells to Be Active

Zearalenone, a non-steroidal mycotoxin [23], has a structure similar to 17 β-estradiol that can compete for binding to the estrogen receptor and affecting uterine development [24]. Studies have shown a dose–dependent relationship between the weight of the mouse uterus and the concentration of ZEN (0~45 mg/kg feed), which has also been confirmed in our study [25]. However, other studies have reported that ZEN greater than 0.3 mg/kg feed leads to a decrease in rat uterine weight [26]. It has become a consensus that ZEN can induce abnormal uterine development. To further explore the effect of ZEN on the uterus, HE staining was performed to observe uterine morphological changes. We found that with the increase in ZEN addition (0~3 mg/kg feed), the endometrial and basal layer thickness increased linearly, the uterine gland density increased, and the endometrial epithelial cells increased. This was confirmed in a study by Zhou et al. (2019) [27]. However, results showing that 20 mg/kg feed of ZEN caused uterine tissue atrophy in rats and promoted cell apoptosis were found in the study by Gao et al. (2008) [28].

Wu et al. (2020) treated endometrial cells with 0~20 μg/mL ZEN for 24 h, and the number of autophagosomes increased with ZEN, leading to autophagy, mitochondrial swelling, severe intracellular vacuolization, and apoptosis in a dose-dependent manner [29]. In addition, the study reported that 20 μmol/L ZEN treated testicular support cells for 24 h and autophagy vesicles with bilayer membrane structure could be observed [30]. In this study, with the increase in dietary ZEN (0~3 mg/kg feed), the nucleolar boundary of the endometrial epithelium was blurred, the ER was seriously vacuolized, and the number of mitochondria increased. Low-dose ZEN can exert estrogen-like and carcinogenic effects and stimulate cell proliferation [31], which may be an important reason for the increased number of mitochondria in the test. Yoon et al. (2019) found that treatment with 5 μmol/mL of ZEN for 24 h would promote the increase in cell activity, while when the ZEN dose was greater than 10 μmol/mg, cell activity decreased [32]. This was also confirmed in the study by Song et al. (2020) [33].

### 3.2. The Pathway of AMPK/mTOR

AMPK is a highly conserved regulator of energy and metabolism, regulated by energy charge (AMP/ATP ratio) and phosphorylated to maintain the homeostasis of energy metabolism during energy stress [34]. The mTOR is an important growth regulator that senses different nutritional and environmental factors [35]. AMPK phosphorylation inhibits the downstream target p70S6K and 4E-binding protein-1 by inhibiting the activity of mTOR. AMPK/mTOR is involved in cell growth, proliferation, and metabolism by regulating translation and protein synthesis [36]. Mei et al. (2021) showed that estrogen 17β-estradiol could promote the expression of SIRT1 and *p*-AMPK, inhibit the expression of *p*-mTOR, induce autophagy, improve cell viability, and promote proliferation [34].

Research showed that 20 μmol/L ZEN led to the expression of AMPK/*p*-AMPK in rat testicular Sertoli cells [37], activated the AMPK/mTOR/P70S6K pathway, induced cell autophagy, and reduced the expression of mTOR/*p*-mTOR [35]. In order to investigate further the mechanism of ZEN-inducing autophagy in the AMPK/mTOR pathway, we conducted transcriptome analysis on porcine uterine endometrial epithelium cells. Moreover, adding a certain dose of ZEN to the weaned piglet’s diet can increase ER, which has been confirmed in our previous study [13]. Results of several key genes in the AMPK/mTOR pathway showed that as ZEN increased (0~40 μmol/L), the relative mRNA expression of ULK1 and TSC2 increased linearly, while Rheb decreased linearly. The binding of ZEN to the estrogen receptor can change the concentration of Ca^2+^ in the cytoplasm, induce phosphorylation of AMPK through phosphorylation of CaMKKβ, then regulate the phosphorylation of TSC2, and inhibit the expression of Rheb. In addition, Rheb is closely related to the activation of mTORC1 (mTOR’s protein complex), and the decreased expression of mTORC1 will reduce the binding of ULK1 to ATG13, promoting autophagy [13,38,39]. Therefore, it is reasonable to speculate that ZEN may mediate autophagy by activating the AMPK/mTOR pathway, but the detailed mechanism still needs further study.

### 3.3. Expression of Autophagy-Related Genes

Autophagy maintains the stability of the internal environment by eliminating toxic substances, damaged organelles, and misfolded proteins [40,41]. If autophagy is inhibited, normal cell function will be affected. When the body is in a stress state, the pathway of AMPK is activated, inducing ULK1 to negatively regulate mTOR [17,18]. ULK complexes can promote the formation of autophagosomes, producing PI3K kinase complexes comprising Beclin1, VPS15, and VPS34, thereby producing phosphatidylethanolamine [42]. When LC3, ATG8, and phosphatidylethanolamine bind to autophagosomes, cellular autophagy is activated [43]. Beclin1 is one of the earliest autophagy-related genes discovered, and it plays a crucial role in the early formation of autophagosomes [44]. Pan et al. (2020) showed that when ZEN was administered orally to rats, the expression of Beclin1 significantly increased when the ZEN was greater than 10 mg/kg feed [45]. In our experiments, the relative expression of Beclin1 in the endometrial epithelium increased linearly with increasing ZEN concentration, consistent with the transcriptomic results of the porcine endometrial epithelial cells in the in vitro model. LC3 is a specific autophagic marker that can reflect the strength of autophagy via both the number of autophagosomes and the level of LC3 lipid formation [21]. Research has shown that with the increase in ZEN, the expression of LC3-II increases linearly [16,46]. Consistent with previous studies, we found that with the increase in ZEN, the relative mRNA expression of LC3 in the uterine endometrial epithelium cell linearly increased. This was also confirmed by the transcriptome mRNA results of in vitro model pig uterine endometrial epithelium cells. ATG5, ATG7, and ATG9 are all key proteins necessary for the formation of autophagosomes. In our experiments, we found that as the ZEN increased, the expression of ATG5, ATG7, and ATG9 in the uterine endometrial epithelium cell increased linearly, and the transcriptome results of porcine endometrial epithelial cells were the same.

### 3.4. Expression of Proliferation and Apoptosis-Related Genes

Cell proliferation autophagy is a self-repairing mechanism and plays a crucial role in the development of eukaryotes [22,47]. PCNA is a marker of cell proliferation [48,49], while BCL2 is a key regulatory factor for cell apoptosis [50]. BCL2 is located on the outer membrane of mitochondria and inhibits cell apoptosis by inhibiting the release of cytochrome c. BAX is a heterodimer of BCL2, which can induce mitochondrial membrane depolarization, increase mitochondrial outer membrane permeability, promote the release of cytochrome c, and eliminate BCL2’s inhibition of cell apoptosis [51,52]. Our experiment showed a positive correlation between the relative protein expression of AMPK and proliferation-related genes and a negative correlation with mTOR.

PCNA plays a crucial role in multiple environments, including cell proliferation, nucleotide and base excision repair, and chromatin assembly, and is a determinant of cell proliferation into the S phase [53]. Similar to previous studies, the results of this experiment showed that with the increase in ZEN, the relative expression of PCNA increased linearly, and when ZEN exceeded 20 μmol/L, the relative expression of PCNA gradually decreased, which may be due to the estrogen-like effect and carcinogenic properties of a low dose of ZEN [31]. The carboxyl terminus of the BCL2 family has hydrophobic gene sequences that can bind tightly to the mitochondrial membrane. The BCL 2 family can activate the mitochondrial apoptotic program when cells are under stress [54]. Research has shown that ZEN will lead to an increase in the relative expression of BAX and a decrease in the relative expression of BCL2 [55]. Similar to previous studies, we found that as the ZEN increased, the relative expression of BCL2 linearly increased while BAX linearly decreased.

## 4. Conclusions

ZEN can activate the phosphorylation of AMPK in endometrial epithelial cells, affect the phosphorylation level of mTOR through TSC2 and Rheb, induce autophagy, upregulate the expression of proliferative genes PCNA and BCL2, downregulate the expression of apoptotic gene BAX, promote endometrial epithelial cell proliferation, and ultimately lead to thickening of the endometrial and muscular layers, increase in glands, and induce uterine hypertrophy.

## 5. Material and Methods

### 5.1. Preparation of ZEN Supplemented Diet

Purified ZEN was purchased from Fermentek (Jerusalem, Israel). The purified ZEN was dissolved in ethyl acetate solution (≥99.5%), and then ZEN was added to the diet and left overnight to evaporate ethyl acetate. All feeds were completed in the same batch and stored in covered containers.

### 5.2. Animals, Experimental Design, and Management

The management and design of the experiment followed animal care rules approved by the Animal Nutrition Research Institute of Shandong Agricultural University and the Ministry of Agriculture of China for the Care and Use of Laboratory Animals.

In the first experiment, a total of 32 female weaned piglets (Duroc × Landrace × Yorkshire, DLY) were randomly allocated to 4 treatments with 8 repetitions per treatment. The experimental diets were basal diets (as described in Appendix A) supplemented with 0 (Control), 0.15 (ZEN0.15), 1.5 (ZEN1.5), and 3.0 mg/kg feed ZEN (ZEN3.0), respectively, and the experimental period lasted for 32 days after the pre-experimental period of 7 days. All weaned piglets were weighed and slaughtered on the last day of the experiment. The uterus was collected, and the histomorphology and ultrastructure of endometrial epithelial cells were determined. In addition, the expression of AMPK/mTOR signaling pathway-related genes, autophagy-related genes, proliferation, and apoptosis-related genes were also determined. In the second experiment, porcine endometrial epithelial cells were resuscitated and cultured. Then, 0 (Control), 5 (ZEN 5), 20 (ZEN 20), and 40 μmol/L (ZEN 40) of ZEN were added in their logarithmic growth phase, respectively. After 24 h of treatment, the cell viability, transcriptome, AMPK/mTOR pathway, and the expression of autophagy, proliferation, and apoptosis-related genes were determined.

### 5.3. Histological Examination

Uterine tissue sections were dewaxed, rehydrated with xylene and graded alcohol concentrations, stained with hematoxylin for 1 min, and rinsed with water until color was no longer evident. The sections were then differentiated, and hematoxylin from the stained cytoplasm was wiped away with hydrochloride alcohol and then incubated in eosin for 1 min. The dehydrated tissue sections were sealed with clear resin.

### 5.4. The Observation of Transmission Electron Microscopy

① Rinse the sample 5 times × 20 min with phosphate buffer, and then fix it with osmic acid for 4.5 h. ② Dehydrate with ethanol, followed by embedding with epoxy ethane I (1 h), epoxy ethane II (1 h), epoxy ethane, resin (1:1; 6 h), pure resin (2 times × 13 h), and pure resin + DMP-30 (15 h). ③ Cut the resin block into 1 μm (RMC POWERTOME-XL, New York, NY, USA). ④ Stain with toluidine blue, select areas with abundant epithelial cells and make ultra-thin sections using an ultra-thin slicer (LKB NOVA, Sollentuna, Sweden). ⑤ The observation was conducted using a transmission electron microscope (JEM-1400 Plus, Tokyo, Japan).

### 5.5. Immunohistochemistry

Sections were processed according to the standard immunohistochemistry (IHC) protocol and then dewaxed, rehydrated, and microwaved at full power for 20 min in sodium citrate buffer (0.01 moL·L^−1^, pH 6.0). Sections were incubated in 10% hydrogen peroxide (H_2_O_2_) for 1.5 h, followed by 1 h in 10% normal goat serum (ZSGB-BIO, Beijing, China). After washing with phosphate-buffered saline (PBS), the sections were incubated with primary antibody at 4 °C. The next day, the sections were washed with PBS and subsequently incubated with copolymer for 1 h at 37 °C. A second incubation of Polink-2 plus HRP anti-rabbit was performed at 37 °C (PV-9001; ZSGB-BIO, Beijing, China), then the sections were washed with PBS and then immersed in tetrachlorinated diaminobenzidine, DAB) using the DAB (TIANGEN PA110, Beijing, China) kit for 1~3 min. The sections were then dehydrated, sealed in transparent resin, and mounted, and the distribution of the immunoreactive substances was observed under a microscope.

### 5.6. Total RNA Extraction, cDNA Preparation, and Quantitative Real-Time Reverse Transcription Polymerase Chain Reaction (qRT-PCR)

Total RNA was extracted from the uterus of prepubertal gilts using RNAiso Plus (Applied TaKaRa, Dalian, China). Total RNA was reverse transcribed to cDNA using the Reverse Transcription (RT) System Kit (PrimeScriptTM RT Master Mix, RR036A, Applied TaKaRa, Dalian, China). We performed a real-time polymerase chain reaction (qRT-PCR) with a total volume of 20 μL (Appendix A). The optimized qRT-PCR protocol was: a 95 °C initial denaturation step for 30 s, i.e., 95 °C denaturation for 5 s, 60 °C denaturation for 34 s, 95 °C denaturation for 15 s, 60 °C denaturation for 60 s, 43 cycles, and the last 95 °C denaturation for 15 s. The qRT-PCR reaction was performed using the AB7500 real-time PCR system (Applied Biosystems, Foster City, CA, USA). The relative quantitative expression of BAX, BCL-2, PCNA, ATG 5, ATG 7, ATG 9, Beclin1, LC3, and β-actin mRNA was calculated to be equal to 2^−ΔΔCT^. The primer sequences and product lengths are shown in Appendix A.

### 5.7. Western Blot Procedure

Total uterine protein was extracted using the lysate instructions (Beyotime, Shanghai, China) and quantified using the bicinchoninic acid (BCA) Protein Assay Kit (Tiangen Biotech, Beijing, China). The sample size was 55 µg of protein per sample. Proteins were separated using polyacrylamide gel electrophoresis and then transferred to a nitrocellulose membrane (Solarbio, Beijing, China). Membranes were incubated in 10% skim dry milk for 2 h, washed three times with tris-buffered saline plus Tween (TBST, pH 7.6), and then incubated overnight with primary antibodies in antibody dilution buffer (Beyotime, Shanghai, China) at 4 °C. The membrane was then incubated with the anti-rabbit IgG (1:2000, Beyotime, Shanghai, China). The buffer with a secondary antibody (Beyotime, Shanghai, China) was diluted at 2.5 h at 37 °C. The membrane was then immersed into a highly sensitive lumen-emitting reagent (BeyoECL Plus, Beyotime, Shanghai, China). We used FusionCapt Advance FX7 (Beijing Oriental Technology Development Co., Ltd., Beijing, China) to expose the film.

### 5.8. Determination of Endometrial Epithelial Cell Activity

The endometrial epithelial cells (PECs) of pigs gifted from the College of Animal Science and Technology, Shandong Agricultural University, were removed from liquid nitrogen and thawed and revived in a 37 °C water bath for 1~1.5 min. PECs have been proven to be applicable in subsequent experiments. After the cells inside the cryopreserved tube are completely melted, transfer the epithelial cells to a centrifuge tube and centrifuge at 1000 rpm for 3 min. Discard the supernatant, add 1 mL of DMEM high glucose medium (containing 10% fetal bovine serum), and blow the cell clusters for 2 min. Finally, add 9 mL of culture medium to the culture dish, evenly distribute the blown cell suspension to various areas of the dish, shake well for 2 min, and place it in a culture incubator (37 °C, 5% CO_2_). Cell viability was measured after the successful culture of uterine endometrial epithelium cells. First, the cells were digested with 0.5% trypsin, and subsequently, 9 mL of complete medium was added, and the cells were resuspended. Then, 1 mL of the suspension was extracted to check the number of cells. The suspensions were inoculated into 96-well plates, and 1 × 10^4^ cells were ensured in the 100 μL cell suspensions per well. Under the microscope, when the number of cells reached 70% of the visual field, the cells were treated with a complete medium containing different concentrations of ZEN. After 24 h and 48 h, a 10 μL CCK solution was added. The cells were incubated at 37° and 5% CO_2_ for 1 h. After incubation, the absorbance at 450 nm was detected using an enzyme marker, and the cell activity was calculated.

### 5.9. Transcriptome Analysis of Endometrial Epithelial Cells

After the cells were treated with ZEN for 24 h, 1 mL of RNAex Pro Reagent solution was added to fully lysate the cells, and the lysate cell suspension was transferred to a 1.5 mL centrifuge tube and left for 5 min. RNA purity and concentration were determined (Denovix DS-11, Wilmington, NC, USA).

The cDNA was sent to Beijing Nowo Zhiyuan Technology Co., Ltd. (Beijing, China) for transcriptome detection. ① Illumina’s NEBNext^®^ UltraTM RNA Library Prep Kit was used to construct the library. ② After the library was obtained, a Qubit2.0 Fluorometer (New England Biolabs, Ipswich, MA, USA) was used for preliminary quantification, and an Agilent 2100 bioanalyzer (Agilent, Palo Alto, CA, USA) was used to detect the insert size of the library after dilution. ③ Illumina sequencing was carried out after pooling different libraries according to the requirements of effective concentration and target data volume, and a 150 bp paired-end reading was generated to obtain the sequence information of the fragments to be tested. ④ Image data were converted into sequence data (reads) by CASAVA base recognition using a high-throughput sequencer containing sequence information of sequenced fragments and corresponding sequencing quality information. The number of reads covered by each gene was calculated according to the location information of genes on the reference genome, and then the feature count of subread software was used to quantify gene expression.

### 5.10. Verification of Relative Protein Expression of Genes

After the cells were treated with ZEN for 24 h, the medium was discarded, washed with pre-cooled PBS 2 to 3 times, and 100 μL of RIPA strong lysate (containing protease inhibitors) was added, placed on ice for cracking for 30 min, and finally transferred to a 1.5 mL centrifuge tube.

### 5.11. Statistical Analysis

The data analysis was conducted using the general linear model (GLM) program of SAS 9.2 (Version 9.2, SAS Institute Inc., Cary, NC, USA) and followed the Shapiro–Wilk evaluation of the normal distribution of the data (*W* > 0.05). One-way ANOVA was used to analyze the differences between treatment groups, and we used the orthogonal polynomial comparison method to perform a linear regression analysis on the treatment effects of different levels of ZEN. We also used the LSD multi-group range test method for multiple comparisons and Pearson correlation analysis to test the correlation differences between each two groups, with a significance level of *p* < 0.05 and an extremely significant level of *p* < 0.01. The data on real-time fluorescence quantitative PCR were analyzed using the 2^−ΔΔCT^ method, and the grayscale values of related protein bands were analyzed using the Image J (×64) (v1.8.0) image analysis software.

## Figures and Tables

**Figure 1 toxins-16-00073-f001:**
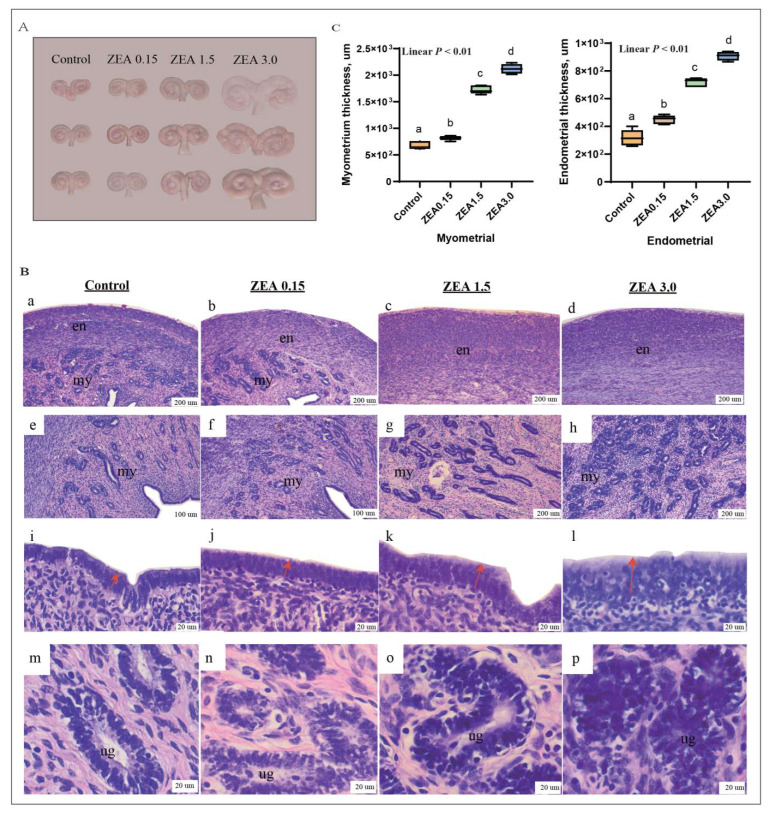
Effects of ZEA on uterine hypertrophy (**A**), histological structure of the uterus (**B**), myometrial and endometrial thickness (**C**) in gilts. The “en”, “my”, and “ug” in (**B**) represent the endometrium, myometrium, and uterine gland, respectively. Control, ZEA0.15, ZEA1.5, and ZEA3.0 represent the control diet with an addition of 0, 0.15, 1.5, and 3.0 mg·kg^−1^ ZEA. In (**B**), **a**, **e**, **i** and **m** are the observation results of the Control group’s uterine myometrium and endometrium at different multiples; **b**, **f**, **j** and **n** are the observation results of the ZEA0.15 group’s uterine myometrium and endometrium at different multiples; **c**, **g**, **k** and **o** are the observation results of the ZEA1.5 group’s uterine myometrium and endometrium at different multiples; **d**, **h**, **l** and **p** are the observation results of the ZEA3.0 group’s uterine myometrium and endometrium at different multiples. In (**C**), values with different letter superscripts mean significant difference (*p* < 0.05), while with same letter superscripts mean no significant difference (*p* > 0.05).

**Figure 2 toxins-16-00073-f002:**
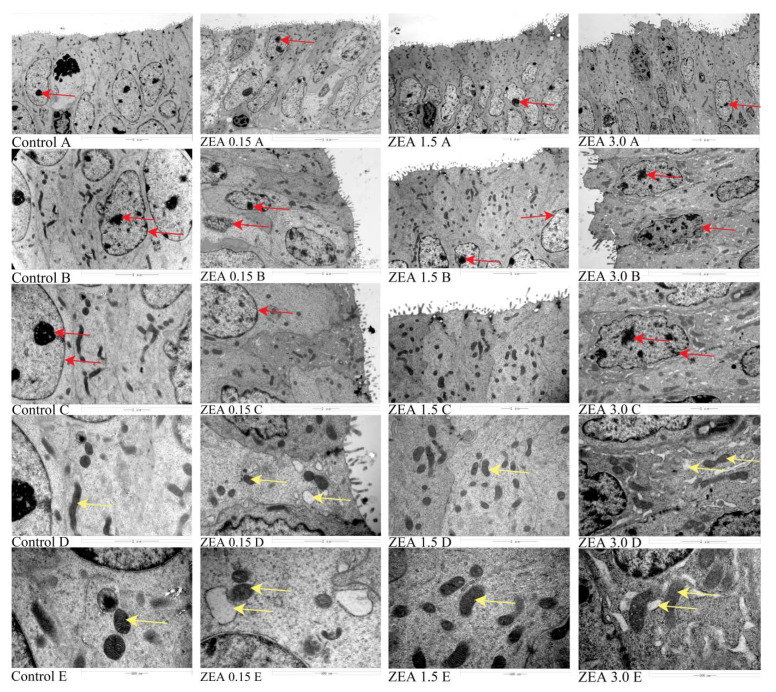
Effects of ZEA on the ultrastructure of uterine endometrial epithelium in gilts. Control, ZEA0.15, ZEA1.5, and ZEA3.0 represent the control diet with an addition of 0, 0.15, 1.5, and 3.0 mg·kg^−1^ ZEA.

**Figure 3 toxins-16-00073-f003:**
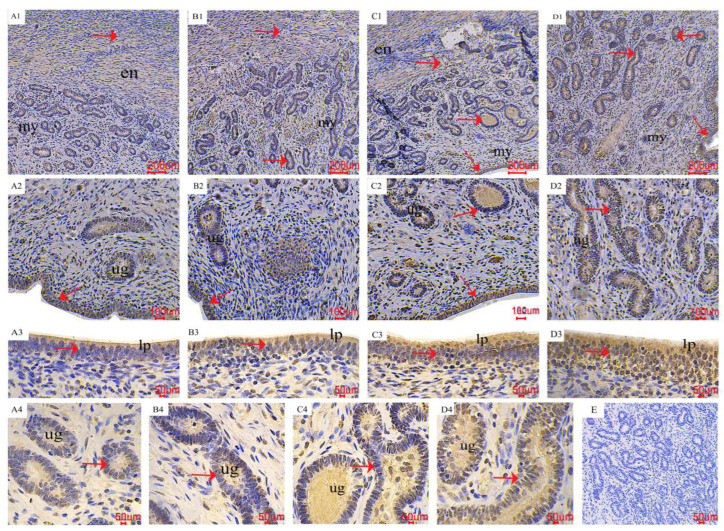
Effects of ZEA on the phospho-AMP-activated protein kinase (*p*-AMPK) localization of the uterus in gilts. The “en”, “my”, “lp”, and “ug” represent the endometrium, myometrium, lamina propria, and uterine gland, respectively. (**A**–**D**) represent the control diet with an addition of 0, 0.15, 1.5, and 3.0 mg·kg^−1^ ZEA. (**E**) was the negative control.

**Figure 4 toxins-16-00073-f004:**
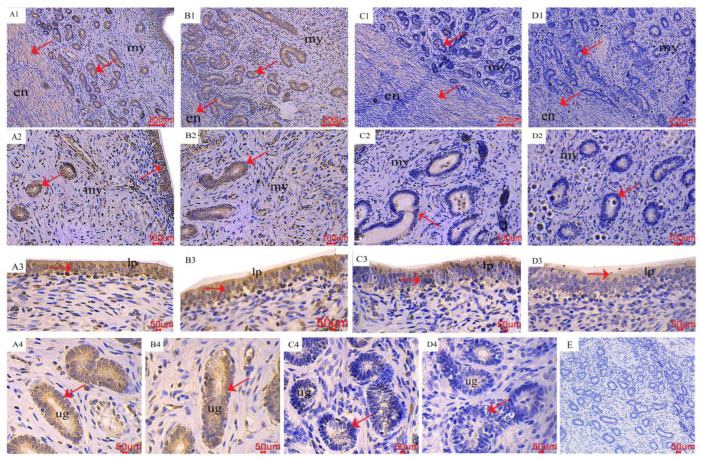
Effects of ZEA on the phospho-mammalian target of rapamycin (*p*-mTOR) localization of the uterus in gilts. The “en”, “my, “lp”, and “ug” represent the endometrium, myometrium, lamina propria, and uterine gland, respectively. (**A**–**D**) represent the control diet with an addition of 0, 0.15, 1.5, and 3.0 mg·kg^−1^ ZEA. (**E**) was the negative control.

**Figure 5 toxins-16-00073-f005:**
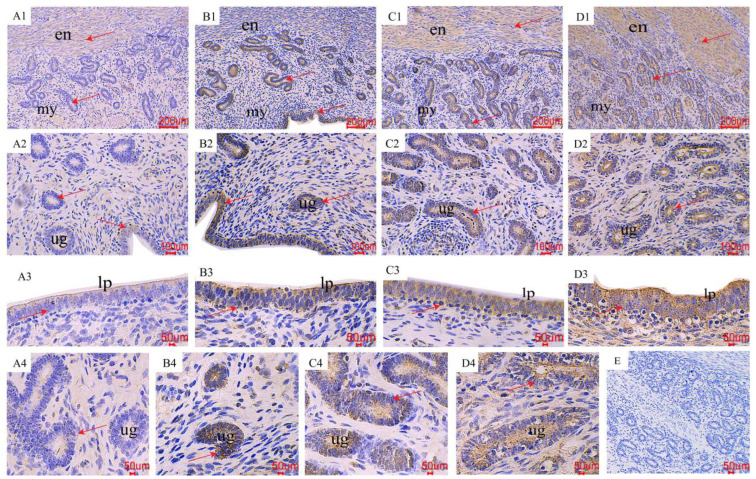
Effects of ZEA on the microtubule-associated protein 1 light chain 3 alpha (LC3) localization of the uterus in gilts. The “en”, “my”, “lp”, and “ug” represent the endometrium, myometrium, lamina propria, and uterine gland, respectively. (**A**–**D**) represent the control diet with an addition of 0, 0.15, 1.5, and 3.0 mg·kg^−1^ ZEA. (**E**) was the negative control.

**Figure 6 toxins-16-00073-f006:**
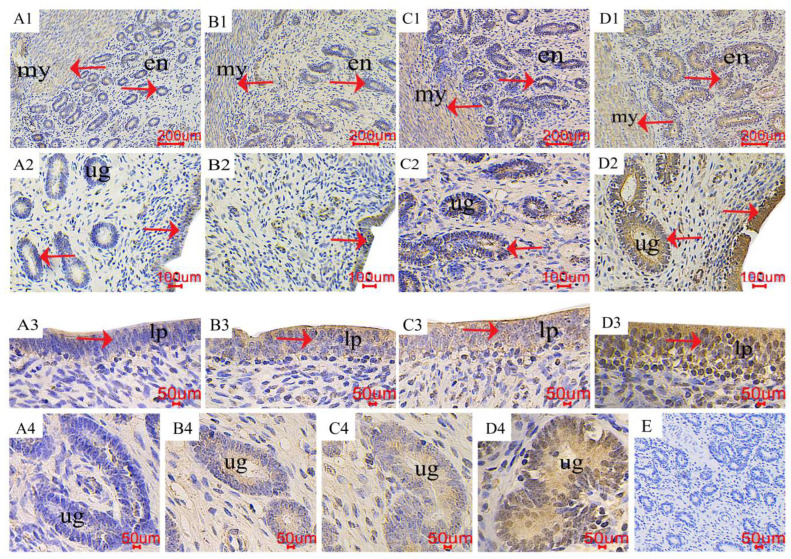
Effects of ZEA on the Beclin1 localization of the uterus in gilts. The “en”, “my”, “lp”, and “ug” represent the endometrium, myometrium, lamina propria, and uterine gland, respectively. (**A**–**D**) represent the control diet with an addition of 0, 0.15, 1.5, and 3.0 mg·kg^−1^ ZEA. (**E**) was the negative control.

**Figure 7 toxins-16-00073-f007:**
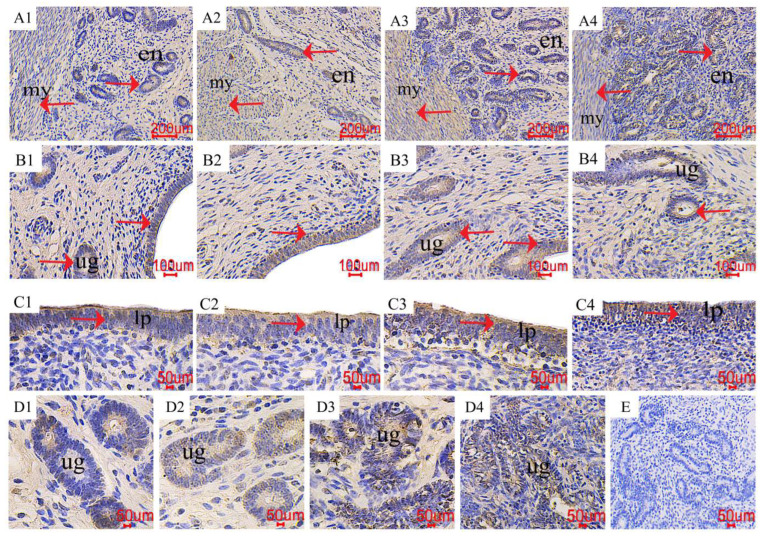
Effects of ZEA on the proliferating cell nuclear antigen (PCNA) localization of the uterus in gilts. The “en”, “my”, “lp”, and “ug” represent the endometrium, myometrium, lamina propria, and uterine gland, respectively. (**A**–**D**) represent the control diet with an addition of 0, 0.15, 1.5, and 3.0 mg·kg^−1^ ZEA. (**E**) was the negative control.

**Figure 8 toxins-16-00073-f008:**
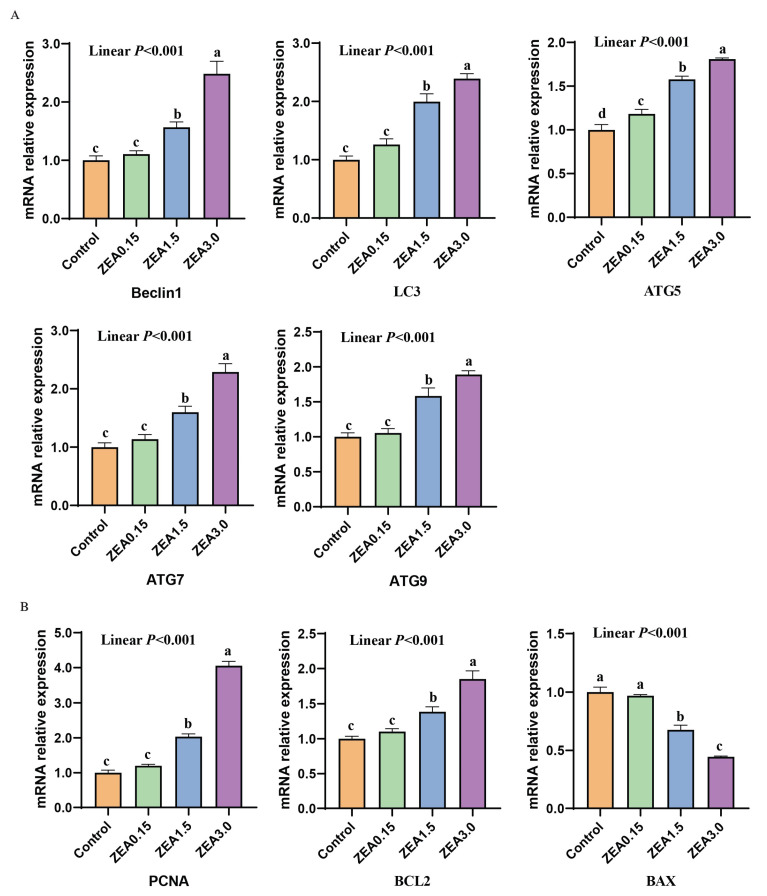
Effect of ZEA on the relative mRNA expression of (**A**) autophagy and (**B**) proliferation apoptosis-related genes. Control, ZEA0.15, ZEA1.5, and ZEA3.0 represent the control diet with an addition of 0, 0.15, 1.5, and 3.0 mg·kg^−1^ ZEA. Values with different letter superscripts mean significant difference (*p* < 0.05), while with same letter superscripts mean no significant difference (*p* > 0.05).

**Figure 9 toxins-16-00073-f009:**
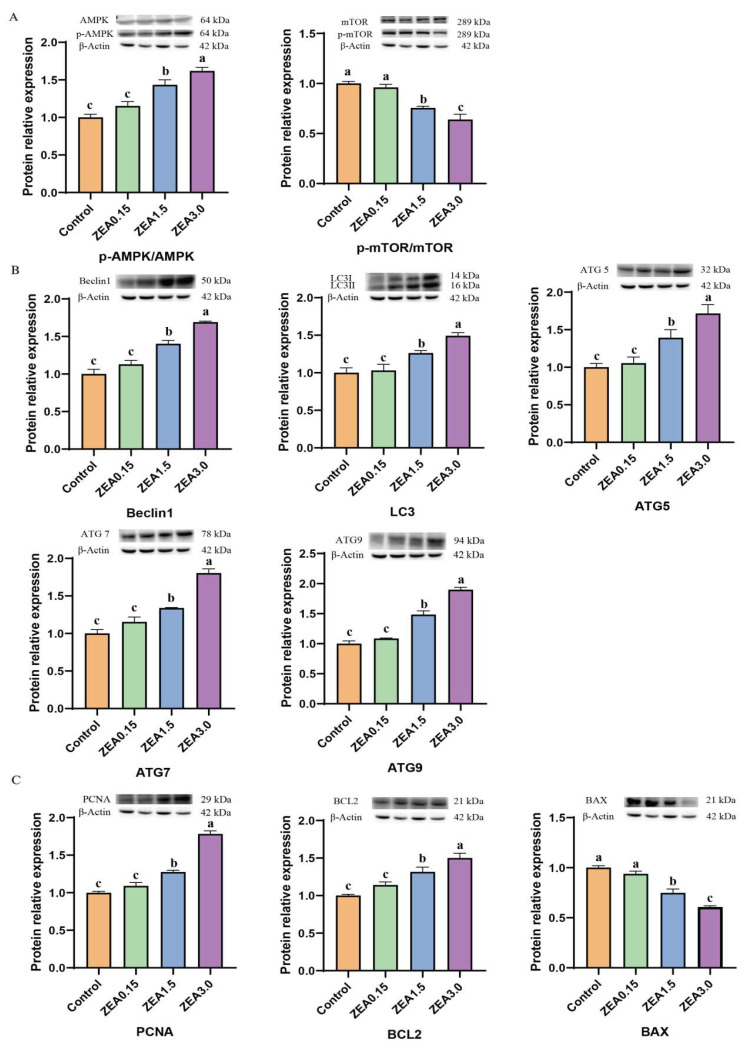
Effect of ZEA on the relative protein expression of (**A**) *p*-AMPK/AMPK and *p*-mTOR/mTOR, (**B**) autophagy, and (**C**) proliferation apoptosis-related genes. Control, ZEA0.15, ZEA1.5, and ZEA3.0 represent the control diet with an addition of 0, 0.15, 1.5, and 3.0 mg·kg^−1^ ZEA. Values with different letter superscripts mean significant difference (*p* < 0.05), while with same letter superscripts mean no significant difference (*p* > 0.05).

**Figure 10 toxins-16-00073-f010:**
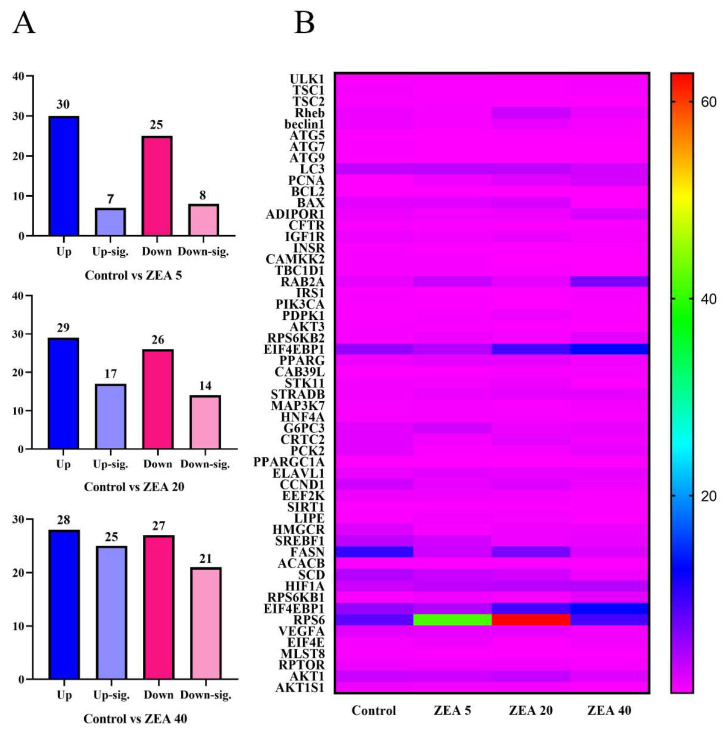
The histogram (**A**) and heatmap analysis (**B**) of zearalenone on AMPK/mTOR pathway-related genes in porcine endometrial epithelial cells. Control, ZEA 5, ZEA 20, and ZEA 40 were porcine endometrial epithelial cells (PECs) exposed to ZEA at 0, 5, 20, and 40 μmol/L for 24 h. Up-sig., significantly upregulated (*p* < 0.05). Down-sig., significantly down-regulated (*p* < 0.05).

## Data Availability

The original contributions presented in the study are included in the article/Appendix A, further inquiries can be directed to the corresponding authors.

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
