# Peer review of "Zearalenone Promotes Uterine Hypertrophy through AMPK/mTOR Mediated Autophagy"

_toxins, 2024, doi:10.3390/toxins16020073_

Round 1
Reviewer 1 Report
Comments and Suggestions for Authors
Comments to authors:
Please note that the official abbreviation for Zearalenone is ZEN (using ZEA is no longer accepted, as is interferes with abbreviation of ZEN metabolites). It would ne appreciated if the authors could add some more recent data on ZEN, occurrence of modified forms, metabolism in pigs to the introduction, as the current text is very generic and not focussed.
For the in vivo experiments a dosage of 0.15 – 5 mg/kg feed was used. Please add the word “feed “in the text to avoid confusion with a dose-range in mg/kg b.w. .
According to the M & M section you started with weaned piglets – and the exposure period was 32 days after a 7-day acclimatization period. In the text the word “gilts” is used. This is not entirely correct; at least you should use the wording “prepubertal gilts” for clarification. Moreover this is important for the interpretation of the ZEN effects, as these animals have at that age no regular oestrus cycle!
Please provide more information about the origin of the porcine endometrial epithelial cells and how these were cultured (cell density, viability testing, morphology where appropriate)
When discussing the dose-dependent effect of ZEN (both in vivo and in vitro) please add to the discussion the agonist/antagonist activity of ZEN on oestrogen receptors to clarify the finding described in 227-239 (and throughout the manuscript). This point of discussion is essential, to enforce the significance of the outlined ZEN- effects, as the ANPK/TOR pathway alterations (as well as autophagy) are common mechanisms that can be induced by various compounds and are not specific to ZN. This part would have been more convincing if the authors had included 17-beta oestradiol and/or Genistein and Daidzeina in their experimental protocol, at least in the in vitro studies. If no additional experiments can be added, such a comparison should be at last included in the discussion section.
The manuscript has a large number of figures, now arranged according to the methods applied (immuno-histology, PCR. The manuscrript would be much more attractive if composite figures could be composed, focussing in the different hypotheses.
Please note that the manuscript, particularly the abstract and introduction require thoroughly editing, as some sentences and expression are incorrect or misleading ( for example: endometrium = uterine endometrial epithelium?; muscular layer = myometrium ?, increase of gland = increased density of uterine glands ?
Line 29: ZEN accumulation? - do you mean “ZEN is distributed to several organs” (accumulation seems not be justified from the toxicokinetic point of view)
Line 69: what is sensorally? (optically?)
Line 27: should read Fusarium graminearum (capital F and entirely in italics)
Comments on the Quality of English Language
This would be very long list - do you have a competent (familiar with the subject) colleague who could do the editing?
Author Response
Reply to Editor and Reviewers
On behalf of my co-authors, we sincerely thank you for giving us an opportunity to revise our manuscript, and deeply appreciate editor and reviewers for their positive and constructive comments and suggestions on our manuscript entitled “Zearalenone promotes uterine hypertrophy through AMPK/mTOR mediated autophagy”.
We have checked the revised manuscript carefully, and look forward to hear from you soon. If you have any questions, please don’t hesitate to let us know. Appended to this letter is our point-by-point response to the comments raised by the reviewers. The comments are reproduced and our responses are given directly afterward in red color.
Reviewer 1:
Please note that the official abbreviation for Zearalenone is ZEN (using ZEA is no longer accepted, as is interferes with abbreviation of ZEN metabolites). It would ne appreciated if the authors could add some more recent data on ZEN, occurrence of modified forms, metabolism in pigs to the introduction, as the current text is very generic and not focussed.
- Thanks for your kind reminder. We found that "ZEA" and "ZEN" are commonly used in the literature in the early stage, so we generally used "ZEA" when writing the manuscript in the early stage. Unfortunately, I only found out today that this is inaccurate. Thank you again for your correction. “ZEA” was changed to “ZEN” according to your suggestion. Also, more recent data and related references on ZEN were added in the section of “Introduction”.
For the in vivo experiments a dosage of 0.15 – 5 mg/kg feed was used. Please add the word “feed “in the text to avoid confusion with a dose-range in mg/kg b.w. .
- Thank you for your kind comments, the word “feed” was added according to your suggestion.
According to the M & M section you started with weaned piglets – and the exposure period was 32 days after a 7-day acclimatization period. In the text the word “gilts” is used. This is not entirely correct; at least you should use the wording “prepubertal gilts” for clarification. Moreover this is important for the interpretation of the ZEN effects, as these animals have at that age no regular oestrus cycle!
- Thank you for your kind reminder. According to your suggestion, the word “gilts” in L107, L578, L583, L589, L595, L601, L607, L614, and L673 was replaced with “prepubertal gilts”.
Please provide more information about the origin of the porcine endometrial epithelial cells and how these were cultured (cell density, viability testing, morphology where appropriate)
- Thank you for your comments. According to you suggestion, necessary information about the origin of the cells was added (The endometrial epithelial cells (PECs) of pigs gifted from the College of Animal Science and Technology, Shandong Agricultural University was removed from liquid nitrogen and thawed and revived in a 37 ℃ water bath for 1~.5 minutes. PECs have been proven to be applicable in subsequent experiments (Zhang et al., 2019). After the cells inside the cryopreserved tube are completely melted, transfer the epithelial cells to a centrifuge tube and centrifuge at 1000 rpm for 3 minutes. Discard the supernatant, pour in 1 mL of DMEM high glucose medium (containing 10% fetal bovine serum), and blow the cell clusters for 2 minutes. Finally, add 9 mL of culture medium to culture dish, evenly distribute the blown cell suspension to various areas of the dish, shake well for 2 minutes, and place it in a culture incubator (37 ℃, 5% CO2).). Also, the related references was added (Zhang K., Li H., Dong S., Liu Y., Wang D., Liu H., Ge L., Su F. and Jiang Y.. Establishment and evaluation of a PRRSV-sensitive porcine endometrial epithelial cell line by transfecting SV40 large T antigen. BMC Vet. Res., 2019, 15(1): 1-10.).
When discussing the dose-dependent effect of ZEN (both in vivo and in vitro) please add to the discussion the agonist/antagonist activity of ZEN on oestrogen receptors to clarify the finding described in 227-239 (and throughout the manuscript). This point of discussion is essential, to enforce the significance of the outlined ZEN- effects, as the ANPK/TOR pathway alterations (as well as autophagy) are common mechanisms that can be induced by various compounds and are not specific to ZN. This part would have been more convincing if the authors had included 17-beta oestradiol and/or Genistein and Daidzeina in their experimental protocol, at least in the in vitro studies. If no additional experiments can be added, such a comparison should be at last included in the discussion section.
- Thank you for your kind comments. Unfortunately, we may not be able to conduct additional test validation of the problem you pointed out, as the project has already been completed and the expensive ZEN may not be immediately available for testing. However, we plan to make a design according to your suggestions in the next study. In addition, we also carry out in-depth comparative analysis in the discussion section. What’s more, adding a certain dose of ZEN to the weaned piglets diet can increase ER, which has been confirmed in our previous study (Yang L J, Zhou M, Huang L B, Yang W R, Yang Z B, Jiang S Z, Ge J S. Zearalenone-promoted follicle growth through modulation of Wnt-1/β-catenin signaling pathway and expression of estrogen receptor genes in ovaries of postweaning piglets[J]. Journal of Agricultural and Food Chemistry, 2018, 66(30): 7899-7906). The main purpose of this study is aims to construct in vivo and in vitro models of ZEN challenge, using AMPK/mTOR as a targeted pathway for ZEN reproductive toxicity, and explore the molecular mechanism by which ZEN may induce uterine hypertrophy in weaned piglets
The manuscript has a large number of figures, now arranged according to the methods applied (immuno-histology, PCR. The manuscrript would be much more attractive if composite figures could be composed, focussing in the different hypotheses.
- Thank you so much for your valuable comments. It should be noted that the method used in this study, especially immunohistology, can perfectly and intuitively show the distribution and accumulation of related proteins to readers by using pictures. Therefore, we have retained some representative pictures in the manuscript to make it easier for readers to understand our results and increase the readability of the manuscript. We feel that pictures seem to grab readers' attention more instantly than numbers.
Please note that the manuscript, particularly the abstract and introduction require thoroughly editing, as some sentences and expression are incorrect or misleading ( for example: endometrium = uterine endometrial epithelium?; muscular layer = myometrium ?, increase of gland = increased density of uterine glands ?
- Thank you for your kind reminder. We apologize for the careless use of vocabulary in our manuscript. Based on your suggestion, we have proofread the language of the entire manuscript. Among them, we have made modifications to the areas you highlighted and highlighted them in red font in our manuscript.
Line 29: ZEN accumulation? - do you mean “ZEN is distributed to several organs” (accumulation seems not be justified from the toxicokinetic point of view)
- Thank you for your comments. We apologize for the careless use of vocabulary in our manuscript. Indeed, the use of "accumulate" here seems unscientific. Upon examination, we feel that the use of "transport" here seems more reasonable.
Line 69: what is sensorally? (optically?)
- Thank you for your kind reminder. Indeed, using "sensorally" to express the results does not seem rigorous. Our intention was to provide a sensory overview of the number of mitochondria, without conducting statistical analysis. Therefore, in order to ensure the scientific nature of the language, we have decided to delete this sentence to avoid confusion for readers.
Line 27: should read Fusarium graminearum (capital F and entirely in italics)
- Thanks for your kind reminder. L13 and L34 in our manuscript were corrected according to your suggestion.

Reviewer 2 Report
Comments and Suggestions for Authors
It is an interesting manuscript where authors have attempted to show uterine hypertrophy in piglets due to zearalenone (Zea), a mycotoxin. The authors explain the induction of autophagy mediated through AMPK/mTOR pathway due to Zea exposure that leads to uterine hypertrophy. Here are some comments which will help in improving the manuscript:
>> The authors have selected three doses Zea 0.15, 1.5 and 3.0 mg/Kg, however it is unexplained why and what basis these doses were selected. Why were these doses given only for 32 days? What is the toxic dose of Zea in animals and humans? Did the authors check how much amount of Zea were present in the tissues/ serum in each treatment group?
>> As the animals were weighed (data not presented in the manuscript), did you see any correlation with uterine tissues with it? Provide animal weight data if you can.
>> It has been mentioned that Zea competitively binds ER to induce abnormal uterine development. The study does not show the status of ER or the estradiol (E2) status which essentially play a major role in uterine development. How does Zea interfere with uterine development. Do you think of an experimental outcome to prove that Zea competes with E2.
>> Line 71-72: Zea 1.5 and Zea 3.0 groups show increased number of mitochondria. Can you explain the reason? How can you correlate its augmented numbers with uterine hypertrophy?
>> Discussion is not compact and crispy. The authors have provided many unnecessary references and enlarged the discussion rather than focusing and discussing their own findings. It needs to be rewritten.
>> Line 256-265, there are repeated sentences. Check it out.
Comments on the Quality of English LanguageEnglish is okay but needs further improvement.
Author Response
Reply to Editor and Reviewers
On behalf of my co-authors, we sincerely thank you for giving us an opportunity to revise our manuscript, and deeply appreciate editor and reviewers for their positive and constructive comments and suggestions on our manuscript entitled “Zearalenone promotes uterine hypertrophy through AMPK/mTOR mediated autophagy”.
We have checked the revised manuscript carefully, and look forward to hear from you soon. If you have any questions, please don’t hesitate to let us know. Appended to this letter is our point-by-point response to the comments raised by the reviewers. The comments are reproduced and our responses are given directly afterward in red color.
Reviewer 2:
It is an interesting manuscript where authors have attempted to show uterine hypertrophy in piglets due to zearalenone (Zea), a mycotoxin. The authors explain the induction of autophagy mediated through AMPK/mTOR pathway due to Zea exposure that leads to uterine hypertrophy. Here are some comments which will help in improving the manuscript:
>> The authors have selected three doses Zea 0.15, 1.5 and 3.0 mg/Kg, however it is unexplained why and what basis these doses were selected. Why were these doses given only for 32 days? What is the toxic dose of Zea in animals and humans? Did the authors check how much amount of Zea were present in the tissues/ serum in each treatment group?
- Thanks for your kind comments. Our team has conducted extensive research on ZEN induced uterine hypertrophy and abnormal ovarian development in weaned piglets. We found that ZEN at 1.0 mg/kg can increase the number of follicles in the ovaries and cause abnormal development. At 0~1.5 mg/kg, there is a dose-dependent increase in uterine volume. According to the 2021 feed hygiene standard: the content of ZEN in formula feed and corn should not exceed 500 μg/kg. The limit of ZEN in the formula feed for piglets is 100 μg/kg. Therefore, this study chose 0.15, 1.5, and 3.0 mg/kg to construct a model of uterine hypertrophy. On the other hand, we also wanted to explore the effects of higher doses of ZEN on uterine development and the expression of related factors. The purpose of controlling the experimental time to 32 days is to ensure that the animals are not affected by their own hormone levels before puberty. Unfortunately, in this experiment, we only measured serum metabolites and toxins in the uterus, and did not measure the levels of ZEN in the blood.
>> As the animals were weighed (data not presented in the manuscript), did you see any correlation with uterine tissues with it? Provide animal weight data if you can.
- Thanks. In this experiment, there was no significant changes in the growth performance of the animals between different treatment groups (Table 3-1). Similarly, we have also conducted a study on the correlation between uterine weight and animal weight, and found that within a certain dose of ZEN, there is a positive correlation between uterine weight and body weight in weaned piglets, but no statistical difference was observed. Therefore, we confirm that uterine enlargement is indeed caused by ZEN.
>> It has been mentioned that Zea competitively binds ER to induce abnormal uterine development. The study does not show the status of ER or the estradiol (E2) status which essentially play a major role in uterine development. How does Zea interfere with uterine development. Do you think of an experimental outcome to prove that Zea competes with E2.
- Thank you for your kind comments. Adding a certain dose of ZEN to the weaned piglets diet can increase ER, which has been confirmed in our previous study (Yang L J, Zhou M, Huang L B, Yang W R, Yang Z B, Jiang S Z, Ge J S. Zearalenone-promoted follicle growth through modulation of Wnt-1/β-catenin signaling pathway and expression of estrogen receptor genes in ovaries of postweaning piglets[J]. Journal of Agricultural and Food Chemistry, 2018, 66(30): 7899-7906). The main purpose of this study is aims to construct in vivo and in vitro models of ZEN challenge, using AMPK/mTOR as a targeted pathway for ZEN reproductive toxicity, and explore the molecular mechanism by which ZEN may induce uterine hypertrophy in weaned piglets.
>> Line 71-72: Zea 1.5 and Zea 3.0 groups show increased number of mitochondria. Can you explain the reason? How can you correlate its augmented numbers with uterine hypertrophy?
- Thank you. As previously reported, there is a close relationship between the enlargement of tissues and organs and energy metabolism. Ma Shijie et al. (2018) found that T2 toxin can increase the number of mitochondria in HEK293T, HepG2, and HeLa cells (Ma S J. T-2 toxin treated with miR-449a/SIRT1/PGC-1 α Pathway induced mitochondrial biosynthesis in animal cells [D]. Wuhan: South China Agricultural University, 2018.). The results of this study indicate that as the concentration of ZEN increases (0~3mg/kg), the nucleolus boundary of the endometrial epithelium blurred, and the number of mitochondria increases. It is speculated that the increase in mitochondrial quantity may be related to cell proliferation and is the intrinsic driving force of uterine hypertrophy. In addition, low-dose ZEA can exert estrogenic and carcinogenic, stimulating cell proliferation (Zheng W., Wang B., Xi. L., Wang T., Zou H., Gu J. H., Yuan Y., Liu X. Z., Bai J. F., Bian J. C. and Liu Z. P.. Zearalenone Promotes Cell Proliferation or Causes Cell Death?. Toxins, 2018, 10(5):184.).
>> Discussion is not compact and crispy. The authors have provided many unnecessary references and enlarged the discussion rather than focusing and discussing their own findings. It needs to be rewritten.
- Thank you for your kind reminder. We have reorganized the section of discussion and highlighted it in red font in our manuscript.
>> Line 256-265, there are repeated sentences. Check it out.
- Thank you for your kind reminder. The repeated sentences were deleted.

Round 2
Reviewer 1 Report
Comments and Suggestions for Authors
The manuscript is greatly improved and the authors addressed all major concerns. In my view the manuscript is ready for acceptance, and only a couple of very minor editorial comments should be addressed in the final processing:
Introduction: delete the 1st sentence – it is an old (and outdated) statement, taken from a recent reference. The manuscript is about ZEN and such general statements are not welcome.
Line 26-28: the (correct) red sentence needs editing (spoken language and incomplete)
Section 3.8. red text: this information is appreciated but the text should be edited to bring in in line with the style of the rest of this section.
Comments on the Quality of English LanguageLine 26-28: the (correct) red sentence needs editing (spoken language and incomplete)
Section 3.8. red text: this information is appreciated but the text should edited to bring in in line with the style of the rest of this section.
Author Response
Reviewer 1:
The manuscript is greatly improved and the authors addressed all major concerns. In my view the manuscript is ready for acceptance, and only a couple of very minor editorial comments should be addressed in the final processing:
Introduction: delete the 1st sentence – it is an old (and outdated) statement, taken from a recent reference. The manuscript is about ZEN and such general statements are not welcome.
- Thank you for your kind reminder. The 1st sentence in the section of “Introduction” was replaced with “Reproductive disorder has become one of the important diseases restricting the high yield of sows. The direct economic losses caused by non-communicable sow reproductive disorders caused by feed mold worldwide each year exceed 100 trillion yuan”.
Line 26-28: the (correct) red sentence needs editing (spoken language and incomplete)
- Thank you for your kind suggestion. The sentence was changed to “ZEN is one of the most common mycotoxins in various cereals of corn, barley, wheat and rice, which is mainly produced by Fusarium graminearum”.
Section 3.8. red text: this information is appreciated but the text should be edited to bring in in line with the style of the rest of this section.
- Thank you for your kind comment. The sentence “After successful culture of uterine endometrial epithelium cells, cell viability was measured. First, the cells were digested with 0.5% trypsin, and subsequently 9 mL of complete medium was added and cell resuspended.” was added to connect the two parts.

Reviewer 2 Report
Comments and Suggestions for Authors
Thanks for the responses to the comments!
Comments on the Quality of English LanguageIt's okay to consider the improved version of the English language.
Author Response
Reviewer 2:
Thanks for the responses to the comments!
- Thank you for your kind comments. On behalf of my co-authors, we sincerely thank you for giving us an opportunity to revise our manuscript, and deeply appreciate you for your positive and constructive comments and suggestions on our manuscript entitled “Zearalenone promotes uterine hypertrophy through AMPK/mTOR mediated autophagy”. If you have any questions, please don’t hesitate to let us know.
